# Coupling and Trapping of Light in Thin-Film Solar Cells Using Modulated Interface Textures

**Jürgen Hüpkes \***[ID]**, Gabrielle C. E. Jost, Tsvetelina Merdzhanova, Jorj I. Owen and Thomas Zimmermann**

Forschungszentrum Jülich GmbH, IEK-5 Photovoltaik, 52425 Jülich, Germany;
jost.gabrielle@googlemail.com (G.C.E.J.); t.merdzhanova@fz-juelich.de (T.M.); jorj.owen@gmail.com (J.I.O.);
thomas_zi@gmx.de (T.Z.)

**\*** Correspondence: j.huepkes@fz-juelich.de; Tel.: +49-2461-61-2594



**Featured Application: Light management is important for solar cell performance. Light-coupling and light-scattering based on rough interfaces are discussed in detail for the example application in silicon thin-film solar cells. The findings are relevant for many photovoltaic technologies.**

**Abstract:** Increasing the efficiency of solar cells relies on light management. This becomes increasingly important for thin-film technologies, but it is also relevant for poorly absorbing semiconductors like silicon. Exemplarily, the performance of a-Si:H/µc-Si:H tandem solar cells strongly depends on the texture of the front and rear contact surfaces. The rear contact interface texture usually results from the front surface texture and the subsequent absorber growth. A well-textured front contact facilitates light-coupling to the solar cell and light-trapping within the device. A variety of differently textured ZnO:Al front contacts were sputter deposited and subsequently texture etched. The optical performance of a-Si:H/µc-Si:H tandem solar cells were evaluated regarding the two effects: light-coupling and light-trapping. A connection between the front contact texture and the two optical effects is demonstrated, specifically, it is shown that both are induced by different texture properties. These findings can be transferred to any solar cell technologies, like copper indium gallium selenide (CIGS) or perovskites, where light management and modifications of surface textures by subsequent film growth have to be considered. A modulated surface texture of the ZnO:Al front contact was realized using two etching steps. Improved light-coupling and light-trapping in silicon thin-film solar cells lead to 12.5% efficiency.

**Keywords:** surface texture; light-trapping; light-coupling; light-scattering; thin-film solar cell; front contact; ZnO:Al

## 1. Introduction

Transparent conductive oxides (TCOs) play an important role as contact materials in solar cells [1,2]. The front contacts simultaneously fulfill numerous requirements: high transparency, high conductivity, and in some cases exhibit a surface texture that facilitates a good optical performance of the solar cell. Many publications have already addressed the importance of balancing the transparency to reduce absorption losses and the electrical properties of TCO films in thin-film solar cells [3–6]. An even larger number of publications address the importance of rough interfaces for optical performance [7–12]. While most studies focus on silicon thin film solar cells, light management has been studied in copper indium gallium selenide (CIGS) and metal halide perovskite technology as well [13–16]. The majority of these publications refer to the necessity of good light-trapping in the device to foster the absorption of the otherwise weakly absorbed red portion of the solar spectrum. Ideally, the light is trapped

completely within the device due to total internal reflection or in other words: it is coupled to waveguide modes [17]. The pathway enhancement [18] is of special importance for the wavelength range that faces a low absorption coefficient (in silicon: 700 nm–1100 nm). Numerous optical models have been designed to describe the scattering effects at the front ZnO:Al/Si interface [19–21]. While the haze, which represents scattering of the rough interface transmitted or reflected into air, is easily measured, scattering of that interface towards the subsequent silicon layer with a higher refractive index reduces the scattering angle by refraction [19]. Recent studies also addressed the importance of the back contact texture rather than the front contact texture [7,22–26]. In a simple picture, the light passes the front contact interface with small scattering angles, while a significant scattering takes place at the rough reflector. However, the coupling to waveguide modes is a complicated interplay between front and back surface texture. Hence, the back contact topography is of great importance for the light-trapping ability of the solar cell. Before light can be trapped it has to enter the absorber layer first. This is facilitated by an improvement of the light-coupling into the device and is very important for technologies using absorbers with high refractive index like silicon. Some groups applied special anti-reflection coatings with intermediate refractive index between the TCO and the silicon [27]. In addition, rough interfaces are well known to reduce reflection losses. An effective refractive index grading at rough interfaces is given once the two materials, e.g., Si and TCO, mesh in the sub-wavelength length scale. The incoming light experiences a gradual change from the pure refractive index of ZnO to the one of Si [28]. The extent of the improvement and the wavelengths that benefit most depend strongly on the texture properties [22]. Small features are required for good light-coupling, while larger features more effectively scatter light. Thus, double textures have been developed including chemical deposition of ZnO [29] or $SnO_2$ [30], etched TCOs [31–33], a combination of those [34,35], and textured substrates [33,36–39]. A theoretical approach to double textured gratings was derived by Lee et al. [40]. Some papers mention the anti-reflection effect [29]. However, a detailed distinction of the effects of light scattering and its anti-reflection effect is lacking.

In this study, we present the analysis of the impact of different surface topographies on the aspects of light-coupling and light-trapping. We utilize sputter-deposited aluminum-doped zinc oxide (ZnO:Al) and subsequent wet-chemical etching to demonstrate various kinds of textures that exhibit different abilities for light-trapping and light-coupling. As a detector, we apply a silicon thin-film tandem solar cell consisting of hydrogenated amorphous and microcrystalline silicon as absorber. Its structure is given in Figure 1a. Based on these findings a double texture is demonstrated, which combines the advantages of providing a light scattering with that of providing a light-coupling interface. The double texture was realized by two-step etching of sputter deposited ZnO:Al. Applied in a tandem solar cell it showed significant improvement in optical performance.

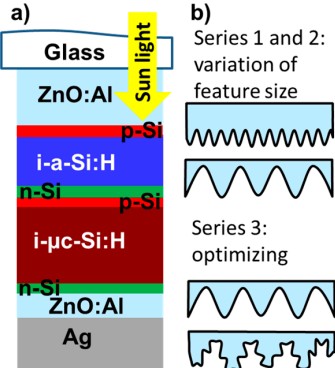

**Figure 1.** Layer stack of the solar cell on glass: aluminum doped zinc oxide front contact, amorphous silicon top cell, microcrystalline bottom cell, and ZnO:Al silver back reflector. The subcells consist of intrinsic silicon absorber sandwitched between p and n doped silicon (**a**), description of test series 1 and 2 for variation of feature size for discussion of the optics and series 3 for optimizing single and double textures (**b**).

## 2. Materials and Methods

### 2.1. ZnO:Al Front Contacts

The ZnO:Al films that were used as front contact layers in this paper were deposited in a vertical inline sputter deposition system (*VISS 300*) by von Ardenne Anlagentechnik, Dresden, Germany. 3.3 mm thick float glass with a size of $10 \times 10$ cm$^2$ (type Eurowhite$^{®®}$ by Euroglas GmbH, Haldensleben, Germany) was used as substrate upon which a 120 nm $SiO_xN_y$ thin-film was deposited via reactive sputter deposition [41] using a Sispa™ target [42,43]. The $SiO_xN_y$-layer served as antireflection coating between the glass and the ZnO:Al and as barrier layer to prevent sodium diffusion from the substrate glass to the zinc oxide interface. The $SiO_xN_y$ deposition conditions were the same for all samples. The ZnO:Al, as well as the SiOxNy depositions, were dual magnetron assisted processes with mid-frequency excitation at 40 kHz. The deposition was conducted in a dynamic mode with the sample carrier moving in front of the target, ensuring a homogenous deposition. More details regarding the deposition system and dynamic deposition process can be found elsewhere [44]. Both processes used argon as sputter gas. The $SiO_xN_y$ was sputtered at 4 kW discharge power using two 100 mm $\times$ 750 mm planar Sispa™ cathodes supplied by W. C. Heraeus GmbH, Hanau, Germany, with a 10 wt. % Al content. Oxygen and nitrogen gas were supplied in the gas phase. The ZnO:Al was sputtered using two 750 mm long rotating tube targets consisting of 0.5 wt. % $Al_2O_3$ in a ceramic ZnO target also supplied by W. C. Heraeus GmbH, Hanau, Germany. The discharge power was constant at 10 kW. The pressure and heater temperature were varied between the individual ZnO:Al depositions to yield different surface textures after etching. Details regarding the process conditions can be found in Table 1. The as-deposited thickness ranged between 880 and 1300 nm with the various deposition conditions. Thicknesses were determined from an etched step using a step profiler without considering volume overestimation of rough surfaces [45]. The sheet resistance was determined using a four-point probe setup. Carrier concentration and carrier mobility were measured by Hall effect measurements in van der Pauw geometry.

**Table 1.** Deposition parameters of the $SiO_xN_y$ and the ZnO:Al layer. $v_{carrier}$ is the velocity of the carrier while passing the target during dynamic deposition.

| Layer | Power Output | Process Pressure | Heater Temperature | Argon Flow | Oxygen Flow | Nitrogen Flow | $v_{carrier}$ | No. of Passes |
|---|---|---|---|---|---|---|---|---|
| | [kW] | [Pa] | [°C] | [sccm] | [sccm] | [sccm] | [mm/s] | |
| $SiO_xN_y$ | 4 | 0.3 | 25 | 200 | 18 | 100 | 3 | 2 |
| ZnO:Al | 10 | 0.3–1 | 430–500 | 200 | - | - | 8 | 7–10 |

### 2.2. Texture Etched ZnO:Al

All ZnO:Al films were textured in a wet-chemical etching step. Three test series were evaluated as illustrated in Figure 1b: test series 1 and 2 consist of single textures with various feature sizes in a wide range, while series 3 was prepared for optimization of single and double textures. The texturing step was performed either in 0.5 w/w% HCl, or 1 w/w% HF, or a combination of double etching processes to yield a broad variety of surface textures [46,47]. Samples in test series 1 were textured in a single HCl (29 s–120 s) or a single HF (180 s) step. Samples in test series 2 were textured in a double etch step with an initial HF etch step (50 s) and a subsequent HCl dip with various durations (0–32 s). The ZnO:Al front contact layers in test series 1 and 2 both feature a simple, single surface texture level. The samples in Section 3.3 (test series 3) on the other hand were textured either in a single HCl etch step or an HCl etch step (29–150 s) followed by an HF dip etch (20–30 s). Details regarding the etching processes with HF and HCl can be found elsewhere [45,48,49]. The two-step etched samples in test series 3 exhibit surface textures with two superimposed texture levels—large and small craters, the so-called double texture.

After etching the sample surfaces were characterized by atomic force microscopy (AFM) and scanning electron microscope (SEM) measurements. In the case of AFM measurements, the 3D topography data were analyzed. A watershed algorithm was applied, to detect the surface features' boundaries and statistically evaluate the average feature diameter. The light-scattering properties of the rough ZnO:Al and silicon/back contact interfaces were analyzed using angular resolved scattering (ARS) measurements of the reflected light to estimate the surface texture evolution during the silicon deposition. This method uses a laser (wavelength 550 nm) to illuminate the surface and detects the distribution of the light scattered into different angles. Though far field measurements do not provide access to the light enhancement in solar cells [17], far field angular intensity distribution (AID) ia a good indicator for the surface topography, because information regarding the roughness, angle distribution or lateral feature diameter influence the measured results [20,50,51].

### 2.3. Solar Cells

After the characterization of the ZnO:Al front contact material hydrogenated amorphous and microcrystalline silicon (a-Si:H/μc-Si:H) tandem solar cells were prepared on these ZnO:Al front contacts. The layer stack is provided in Figure 1a. a-Si:H has a bandgap of around 1.7 eV and is used in the top cell, while μc-Si:H with an indirect bandgap similar to crystalline silicon (~1.1 eV) is used as the bottom cell absorber. All cells were stacked as p-i-n/p-i-n tandem cells without the application of an intermediate reflector between the top and bottom cells. The p-i-n refers to the doping p-type (as hole contact), intrinsic (absorber), n-type (electron selective contact) and all layers consist of a-Si:H or μc-Si:H [52]. The tandem solar cells of test series 1 and 3 were deposited using a single-chamber plasma-enhanced, chemical vapor deposition (PECVD) process in a vacuum deposition system by von Ardenne Anlagentechnik, Dresden, Germany. Details regarding the single-chamber deposition process and the deposition system, in general, can be found in [53]. All samples of test series 1 were co-deposited and yielded a total absorber thickness of ca. 1.9 μm (ca. 320 nm i-a-Si:H top cell, ca. 1.56 μm i-μc-Si:H bottom cell, ca. 80 nm doped layers). The thickness values given here are only approximate values for each test series because the solar cell thicknesses may vary slightly from substrate to substrate even within the same silicon deposition [54]. All samples of test series 3 were likewise co-deposited, however, with a reduced bottom cell thickness, resulting in a total absorber thickness of ca. 1.6 μm (ca. 320 nm i-a-Si:H top cell, ca. 1.25 μm i-μc-Si:H bottom cell, ca. 80 nm doped layers). The absorber layers of test series 2, on the other hand, were co-deposited in a large area PECVD system by Materials Research Group, Wheat Ridge, Colorado, USA. The total absorber thickness of test series 2 was ca. 1.8 μm (ca. 330 nm i-a-Si:H top cell, ca. 1.44 μm i-μc-Si:H bottom cell, ca. 80 nm doped layers). All tandem solar cells feature a ZnO:Al/Ag back contact that was deposited via sputter deposition in the previously mentioned sputter coater. The cell area used for evaluations is $10 \times 10$ mm$^2$ and was defined by a laser scribing process of the P3 line [55]. The IV-characteristics of all solar cells were measured under illumination with an AM1.5g spectrum at standard test conditions using a class AAA sun simulator. Additionally, the external quantum efficiency (EQE) was determined using a spectral response measurement based on 22 supporting points in the wavelength range between 350 nm and 1100 nm. The wavelength selection was performed using band pass filters and the sun simulator spectrum. From these measurements the short-circuit current densities of the top and bottom cells were calculated using the AM1.5g solar spectrum. The crystalline volume fraction of the microcrystalline silicon films was determined by RAMAN scattering [56].

## 3. Results

### 3.1. Solar Cells on Single Textured ZnO:Al with Different Feature Size

All ZnO:Al samples were deposited within the chosen deposition regime in Table 1 and exhibit carrier concentrations in the range of $2.8$–$3.3 \times 10^{20}$ cm$^{-3}$ and carrier mobilities in the range of 25–45 cm$^2$V$^{-1}$s$^{-1}$. The sheet resistance of all films in test series 1 and 2 range between 4.8 and 21.5 Ω. The sheet resistance

is low enough for good electrical performance of thin-film silicon tandem solar cells. The spread in electrical properties and slightly different film thicknesses might cause minor effects of parasitic free carrier absorption in the long wavelength range [5]. In Figures 2 and 3 the short-circuit current densities (calculated using EQE measurement) of the a-Si:H top cell (a), μc-Si:H bottom cell (b), and the root-mean square roughness ($R_q$) of the ZnO:Al front contact texture (c) are displayed as a function of the average feature diameter ($d_{ave}$) on the ZnO:Al surface for series 1 and 2, respectively. The ZnO:Al films of series 1 were prepared at various deposition conditions and have been etched either in HF only (shaded area) or in HCl by single step etching. The HF etched ZnO:Al films show a significantly smaller and less spread average feature diameter than samples etched in HCl. The different surface textures cause variations in optical solar cell performance. All HCl etched samples show almost the same short-circuit current density in the top cell whereas the short-circuit current density in the bottom cell increases with increasing average feature diameter. It is noted that for all HCl etched ZnO:Al samples the roughness increases with increasing feature diameter as well. The HF etched samples, on the other hand, show the smallest average feature diameter, however, due to the steeper crater angles some of these samples exhibited higher roughness than the HCl etched samples with slightly larger feature diameter. The roughness in conjunction with the small feature diameter leads to an increase in top cell short-circuit current density. The bottom cell current, however, does not reach the maximum value of the HCl etched sample. While small features around 400 nm with high roughness exceeding 90 nm boost the short-circuit current density in both top and bottom cell, larger features (ca. 850 nm) with roughness above 90 nm only improve the bottom cell short-circuit current density.

The second test series is used to extend the investigated parameter regime of lateral feature diameter and roughness further (see Figure 3). The surface textures were created on the same ZnO:Al base material. Using an initial HF etching step—50 s for all samples—numerous points of attack were generated on the ZnO:Al surface. A subsequent dip in HCl then widened the craters [46]. With this method, a variety of different average feature diameters can be fabricatedfabricated.

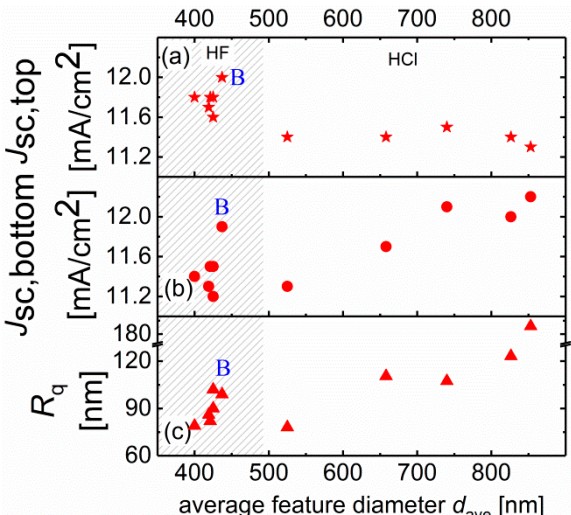

**Figure 2.** Short-circuit current density calculated from external quantum efficiency measurements of (**a**) top cell Jsc, top, (**b**) bottom cell Jsc, bottom, and (**c**) rms-roughness Rq as a function of the average feature diameter of textured ZnO:Al surface for test series 1. The sample 'B' is further discussed in Section 3.2.

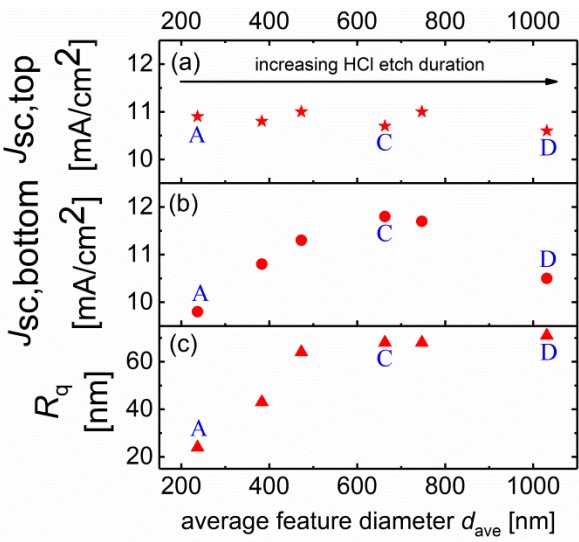

**Figure 3.** Short-circuit current density calculated from external quantum efficiency measurements of (**a**) top cell Jsc, top, (**b**) bottom cell Jsc, bottom and (**c**) rms-roughness Rq as a function of the average feature diameter of textured ZnO:Al surface for test series 1. The samples 'A', 'C', and 'D' are further discussed in Section 3.2.

In comparison with test series 1, the roughness values of test series 2 are significantly lower. Especially the textures with small feature sizes (200 nm–400 nm) exhibit roughness values below 60 nm. The roughness values of samples with larger feature sizes, however, saturated at about 70 nm. The solar cells were deposited in a different deposition system, so the quantitative current density values are not directly comparable to series 1. The short-circuit current density of the top cells does not show any significant variation with the surface texture. The maximum in bottom cell current density was observed at feature sizes of about 700 nm, which is slightly smaller than in test series 1.

Based on the deviations in solar cell performance between test series 1 and 2 it can be concluded that not only the feature size but also the roughness of the texture and therefore the depth of the features significantly influence the optical performance [12,57]. In order to interpret the function of different texture features, it is necessary to understand the two basic effects of enhancing the optical performance of a solar cell: light-coupling and light-trapping.

Improved light-coupling causes the light to enter the solar cell efficiently by decreasing reflection at the interface between ZnO:Al and silicon. Here, the reduced reflection is achieved by an effective index grading at the rough ZnO:Al/silicon interface. The second optical effect, light-trapping, is caused to a large extend by light scattering at the rough reflector.

The texture of the back reflector in superstrate technology is mainly influenced by two factors: the front contact texture and the evolution of the growing absorber layer's surface. This modification is governed by a flattening of the rough interface and by the growth inherent texture of the absorber layers [21,58–60]. Studies have shown that the growth of silicon is non-conformal [21,58,61]. The original front contact texture is overgrown by a cauliflower-like structure in which the dimensions scale with the deposited film thickness.

Keeping this in mind the trends in short-circuit current density that was measured in relation to the average feature diameter and the roughness of the textured ZnO:Al surface can be attributed to the afore mentioned optical effects light-coupling and light-trapping. Features of around 400 nm in diameter with a significant roughness (HF etching, series 1) represent a good effective medium and induce better light-coupling into the solar cell. Figure 4 shows cell reflection measurements on HF and HCl etched samples from series 1. The cell reflection is visibly reduced in the short wavelength region using the HF etched samples as compared to the HCl etched samples. The effect will also affect the long wavelength range, but it is hidden behind the light-trapping effect. Therefore, the current of top

and bottom cells are both increased. The strong interference fringes in the near-infrared region are caused by the low absorption coefficient of the silicon (below 100 cm$^{-1}$ at 1000 nm). The amplitude of the interference is reduced for effectively scattering interfaces.

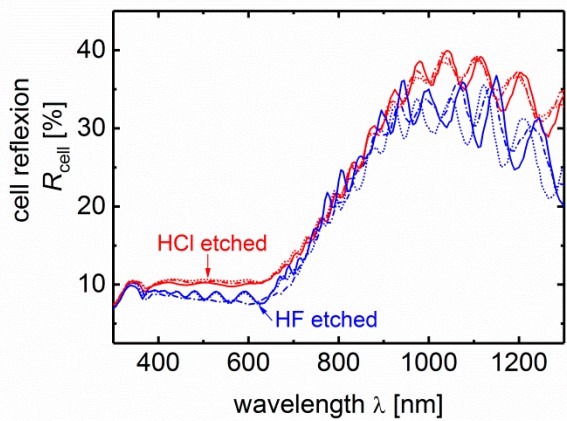

**Figure 4.** Cell reflection measurements of various HF (blue lines) and HCl (red lines) etched samples from series 1.

Figure 5 shows the external quantum efficiency (EQE) of the top and bottom cells of an HF and HCl etched ZnO:Al sample from test series 1. In addition, we provide optical absorption spectra for a typical ZnO:Al front contact and top and bottom cell absorber calculated from absorption coefficients. Note that for the bottom cell we calculated the absorption for twice the absorber layer thickness given the light is absorbed a second time after reflection at the rear side. The strong absorption enhancement in near infrared spectrum in the solar cells proves significant light-trapping for both types of texture.

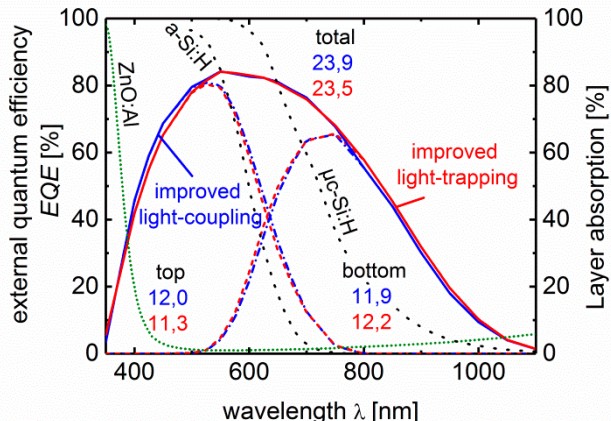

**Figure 5.** External quantum efficiency (EQE) of an HF etched sample with improved light-coupling (blue line) and HCl etched sample with improved light-trapping (red line). The calculated top and bottom cell short-circuit current densities are indicated. The dotted lines represent absorption of typical layers calculated from absorption coefficients given in literature [52] for 1000 nm ZnO:Al (short dotted), 320 nm a-Si:H, and 3120 nm μc-Si:H, assuming a single path absorption for top cell and path length enhancement of 2 utilizing the back reflector for the bottom cell absorber. The calculations neglect any light trapping, reflection or parasitic absorption losses in other layers.

The HF etched ZnO:Al leads to an improved light-coupling due to a refractive index grading (blue line). The increase in the visible range of the a-Si:H cell especially indicates that more light enters the solar cell. This part of the spectrum is generally absorbed very efficiently and does not rely on any pathway elongation. Thus it is mainly governed by the light-coupling.

The reduction in primary cell reflection would also improve the bottom cell quantum efficiency in the NIR spectral range. However, the low NIR absorption in silicon and reflection at the rear reflector result in an additional contribution to reflected light and reflection does not provide sufficient information.

The enhanced EQE in the NIR proves increased absorption from the light-trapping effect induced by the large features etched with HCL. Thus, the two effects light-coupling and -trapping can be distinguished quite clearly here. The maximum value in the short-circuit current density of the bottom cell can be attributed to a better light-trapping effect.

In test series 1, the front contact features are larger (ca. 850 nm lateral) and show a significant depth (rms roughness around 120 nm). Therefore, these structures cannot be smoothened completely by the non-conformal silicon growth and the back reflector still shows a significant texture. In Figure 5 the EQE of the tandem cell with the highest bottom cell current density in test series 1 is displayed (red line). The increase in current in the bottom cell occurs especially in the NIR portion of the light, which strongly relies on a good light-trapping mechanism. The top cell current, on the other hand, shows a strong reduction in the short-circuit current density in comparison to the HF etched surface (blue line) which shows good light coupling properties. In addition to these effects there is a slight shift from the top cell to the bottom cell short-circuit current density in the transition region around 650 nm when changing from HF to HCl etched which also contributes to the differences in the top and bottom cell current densities. This could be due to a minor difference in cell thickness and/or the different scattering behavior at the front contact.

Since the presented test series involved variously deposited and etched samples there might be additional effects involved influencing the absolute optical performance of each front contact within the tandem cell devices. The individual carrier density and film thickness after texture etching and the different back reflector textures may lead to variations in parasitic absorption. However, this mainly affects the bottom cells due to the excellent transparency of all ZnO:Al films in the visible spectral range. Furthermore the crystallinity of the μc-Si:H bottom cell was affected by different front contact textures although the amorphous top cell will alleviate the effects of the front contact texture on the bottom cell [62]. In the bottom cells of test series 2, the crystalline volume fraction as determined by Raman scattering measurements deviates between 39.6–53.6%. Therefore the bottom cells can exhibit different absorption coefficients although the samples were processed within the same silicon deposition. The reduction of reflection losses are attributed to the textured ZnO:Al/silicon interface. In addition, the HF etching of glass might lead to rough features or modify the glass morphology [63]. This could reduce reflection at the glass/air interface. However, we have never observed any change in glass haze or reflection, which is consistent with the low concentration of the hydrofluoric acid and the short etching duration. Thus, we completely neglected this effect.

*3.2. Light Trapping Model*

In the following we will provide an explanation of these effects based on the anti-reflection effect by the effective medium approximation at the rough front interface, light scattering mainly attributed to reflecting interfaces and simple ray tracing.

Figure 6 displays four key surface textures that were previously indicated in Figures 2 and 3 by the letters A through D, to explain the interrelation between surface texture and the optical effects of coupling and trapping in the solar cells. The light-coupling effect is discussed based on the front contact texture only whereas the evaluation of the light-trapping effect also involves the rear surface texture after silicon growth because a significant scattering event occurs here [24]. Thus, the interplay between the two interface textures influences the results. The arrows shown besides the labels "coupling" and "trapping" within Figure 6 indicate the ability of the respective interface to improve light-coupling and –trapping, respectively. Arrows directed upward, to the right, and downward indicate strong, medium and poor abilities.

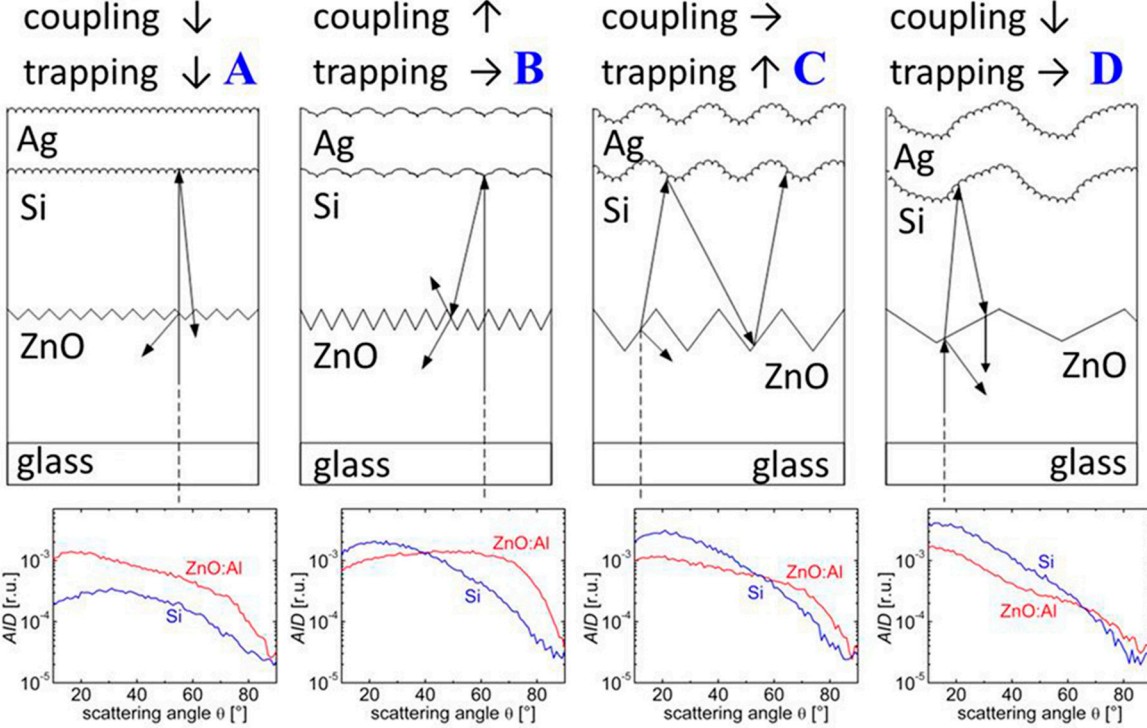

**Figure 6.** Schematic overview of different texture types and their effects on light-trapping and light-coupling. The angular resolved scattering measured at the ZnO:Al/air-interface (red lines) and at the silicon/air- interface (blue lines) is plotted in the graph below each schematic.

Surface A (test series 2) shows small features (<250 nm) in diameter with low roughness (<25 nm). This type of surface exhibits features with lateral feature diameters similar to surface B which shows good light coupling properties, however unlike surface B its craters are lacking sufficient depth to generate an antireflective refractive index grading. Additionally, these small features are flattened by the silicon growth [21,59] preventing significant light-scattering at the back surface of the cell. Hence, there is not any noteworthy light-trapping effect.

This flattening effect can also be seen by the angular resolved reflection measurement that was performed on the rough ZnO:Al-air interfaces prior to cell deposition and on the silicon-air interface at the cell's rear side. Note that the absolute values cannot be compared easily as the measurements on ZnO:Al and on Si yield different reflectivity values due to different refractive indices at the probe wavelength (ZnO: ca. 1.9, Si: ca. 3.8). The scattered intensity diminishes when comparing the ZnO:Al surface and the evolved silicon surface. The different refractive index of ZnO:Al and silicon would suggest the contrary effect. Hence, the very poor scattering intensity is a result of the smoothing effect. Therefore, surface A does neither support light-coupling nor light-trapping as a result of light scattering at the back reflector.

The type B surface (test series 1) on the other hand shows features sizes around 400 nm with a roughness around 90 nm. This surface type facilitates good light-coupling due to an effective refractive index grading. At the same time, the features are not totally smoothed out by silicon growth due to their vertical expansions. Therefore, this surface type also exhibits some light-trapping effects. The strong large angle scattering of the ZnO:Al interface is caused by the significant roughness of the small features, but the scattering angles shift due to the smoothening effect by the silicon growth. The overall scattering intensity, however, remains almost constant.

The type C surface (test series 2) shows larger feature sizes around 700 nm–1000 nm in diameter and a roughness value around 70 nm. In this case, the maximum light-trapping in our test series was achieved. The large surface features are passed to the rear side of the solar cell with only minor modifications by the silicon growth. The AID reveals a reduction of the intensity towards large

scattering angles, but it is less pronounced as for previous cases A and B. Therefore, the incident light beam strikes a rough surface on the rear side, which results in good light-trapping. Although the front contact type C shows roughness values of 70 nm and is therefore within the range of sample B the light coupling in the visible wavelength range is less effective. The larger lateral feature size exceeds the wavelength of the light, so the requirements for the effective medium theory are not fulfilled [28].

The type D surface (test series 2) shows average feature diameters beyond 1000 nm, while its roughness is similar to surface C. This leads to an even less effective light-coupling but also to a reduction in the light-trapping effect. The AID measured in reflection indicates, that the surface angles are passed to the back side almost unchanged, the scattering to large angles decreases slightly. Given the dimensions of these surface structures, we are entering a regime where the cell thickness and the lateral dimension of the structures are similar. If we consider the wavelength in silicon, geometric optics provides a reasonable picture [51]. Light, that enters the tandem solar cell with a type D front contact, is likely to be slightly tilted in its path by diffraction at the front contact. Due to the vast lateral dimensions of the craters, the reflected light beam may strike the front surface on the same facet as the incident one. Therefore, the light beam is easily released to the outside world rather than being trapped inside the solar cell by a large number of random scattering events.

The model simplified the description of the optics in the thin film solar cell that can explain the observed effects. However, a detailed description of coupling to the waveguide modes and the interaction of the rough interfaces with often applied intermediate reflectors requires detailed simulations [17,64,65].

It was shown that the two optical effects of light-coupling and light-trapping are caused by different texture features. The results of test series 1 and 2 also demonstrate that the two different absorber materials in a tandem solar cell are impacted to a different extent by these two effects. Good light-trapping can enhance the optical performance of the μc-Si:H bottom cell significantly but does not necessarily influence the a-Si:H top cell. The interplay between rough interfaces and intermediate reflector layers is discussed in literature [64–66]. A good light-coupling, on the other hand, increases the top and the bottom cell current. However, the bottom cell current is still enhanced more significantly by light-trapping structures. Therefore, it is favorable to combine these two effects with each other to generate the maximum power output in the tandem device. It was already demonstrated that two types of textures are needed to enable the two effects. Thus, a single texture cannot efficiently facilitate both effects simultaneously.

### 3.3. Good Coupling and Scattering Using Double Textured ZnO:Al

A double texture is needed to generate a suitable texture for maximum current output. According to the previously discussed test series 1 and 2, this texture should exhibit large features with significant depth to facilitate good light-trapping at the back reflector and smaller very sharp features to enable a good light-coupling at the same time. Note, that high aspect ratios cause an area increase of the contact interface, which results in stronger interface recombination [67], and sharp features may deteriorate the quality of the absorber layer [68]. A tradeoff between good optical and good electrical device performance must be found.

One possibility to generate a double structure with sputter deposited ZnO:Al is the use of a double etching step with two acids. It was already demonstrated that etching in HF in comparison to etching in HCl shows more points of attack on the same base material and leads to sharper surface features. Using a double etching step in first HCl to define the primary texture with larger features and second HF to define the secondary texture with smaller sharper features shows the possibility to generate a double-textured light-trapping/coupling concept. The ZnO:Al film properties, that are controlled by the deposition conditions, influence the texturing outcome of both etching steps. Therefore, lateral and vertical dimensions of the etch features had been adjusted by the deposition and by the two etching processes [69]. The ranges of the deposition parameters are given in Table 1.

The sputtered ZnO:Al front contacts were optimized for tandem solar cell application using a high rate sputtering process on float glass. Figure 7 shows two SEM surface images of the optimized ZnO:Al single texture after the first etching step in HCl (120 s, left) and the optimized double texture after the second etching step in HF (20 s). For optimization, both deposition parameters and etching parameters were adjusted in test series 3. During the second etching step, small craters have formed on top of the primary HCl-etched texture. There are reports in the literature that further demonstrate the double texture type of surface [64,70]. The textures shown here are realized with high rate sputtering processes onto industrial float glass. The presented single- and double-textured ZnO:Al films were optimized regarding their optical performance in tandem cells. The size and density of small craters in the secondary texture varied between the different ZnO:Al depositions. The deposition and etching conditions had to be adjusted to yield a trade-off between good HCl etching conditions ensuring a suitable crater size for light-trapping and good HF etching conditions ensuring the formation of suitable smaller craters for light-coupling.

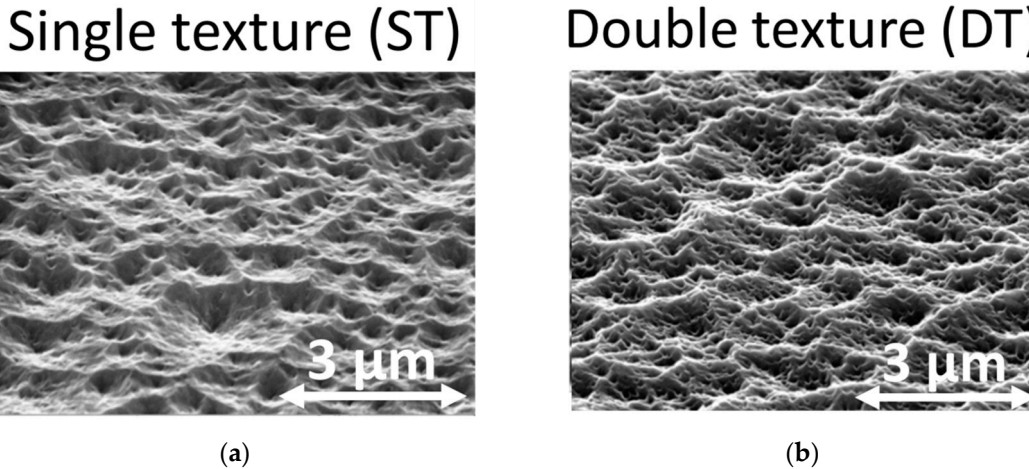

**Figure 7.** SEM surface images of single-textured (ST) and double-textured (DT) surfaces: single-textured after HCl etch only (**a**), double-textured with HCl etch + subsequent HF-dip (**b**).

Figure 8 shows the external quantum efficiency (EQE) and the cell reflectance values of the single-textured sample (ST, blue line) and the double textured (DT, red) sample in Figure 7. The short-circuit current densities calculated using the EQE of the top and bottom cells are given in the graph. By applying the double texture an increased short-circuit current density for both top and bottom cell is observed. In this case the increase in top cell current is more pronounced with 0.8 mA/cm$^2$ as compared to an increase of 0.3 mA/cm$^2$ in the bottom cell. The cell reflectance measurement clearly shows less cell reflection and hence better light coupling with the double-textured front contact in the visible range. As the second etching step applied for the double texture reduces the front contact film thickness, less NIR absorption occurs in the front contact ZnO:Al layer increasing the bottom cell current. In this particular case, the enhancement of the bottom cell current is attributed to lower parasitic absorption in the front contact rather than better light-trapping. The combined top and bottom cell current increased by 1.1 mA/cm$^2$ on the double textured ZnO:Al film as compared to the single texture ZnO:Al film. Using the double texture, the best tandem solar cell yielded an initial efficiency of 12.5% (FF: 76.5%, $V_{oc}$: 1.39 V and $J_{sc}$ = 11.7 mA/cm$^2$) in the remarkably thin total silicon absorber thickness of only 1.64 μm (measured on the particular sample). The two absorber layers and all the doped silicon layers had been deposited in a single chamber process. As reference the best tandem solar cell on the single-textured sample yielded an initial efficiency of 12.1% (FF: 75.7%, $V_{oc}$: 1.40 V and $J_{sc}$ = 11.4 mA/cm$^2$). Thus, the concept of a double texture to increase the short-circuit current density has proven to be very efficient without deterioration of electrical performance. The double-textured (DT) ZnO:Al was additionally included in a comparative study of front contact materials with other

state-of-the-art TCOs showing superior optical performance in a-Si:H/μc-Si:H thin-film solar cells with an accumulated short-circuit current density calculated using EQE measurements of 29.5 mA/cm$^2$ [71].

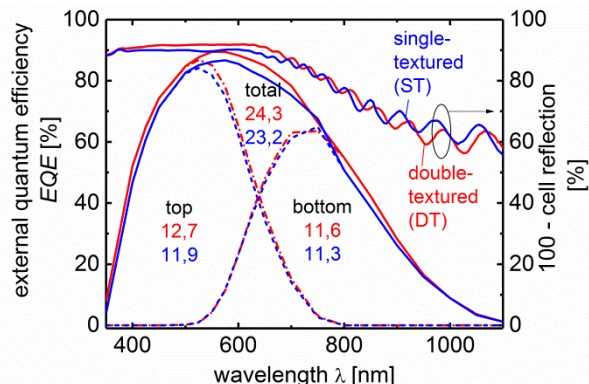

**Figure 8.** EQE and cell absorption (100—cell reflection) of single-textured (ST) HCl etched sample (blue lines) and double-textured (DT) HCl + subsequent HF-dip with improved light-coupling (red lines). The calculated top and bottom cell short-circuit current-densities [mA/cm$^2$] are indicated in the graph. Best cell on each substrate shown respectively.

The described effects on light-coupling and light-trapping are based on the interface textures at the front and rear interfaces. In most cases, one of these interfaces is engineered and the other one just evolves during absorber layer growth [21,59]. Since light management recently has been adopted by other thin film technologies, the analysis of light-coupling and -trapping is of general interest and should be considered in future solar cell development [13–15].

## 4. Conclusions and Outlook

We investigated the impact of different front contact topography types on the optical effects of light-coupling and light-trapping in solar cells. As a well-established example, we demonstrate these effects in a-Si:H/μc-Si:H tandem solar cells. Here light-trapping and light-coupling are very important due to the low absorption of the NIR light and the high reflectivity of silicon with its high refractive index, respectively. The average feature diameters and roughness of the textured front contacts were characterized. Single textures with medium size features (700 nm–1000 nm) and roughness values above 80 nm yielded the best light-trapping. The back reflector interface texture plays a major role for light-trapping. Front contact features with small lateral or vertical size are effectively smoothed by the growth of the absorber layer while large lateral features of the order of absorber layer thickness become effectively flat without relevant light scattering. In contrast, smaller features (ca. 400 nm) with significant roughness around 100 nm enabled a good light-coupling thus. The maximum current density is generated from a solar cell when these two effects are combined in a single solar cell. Double textures made by a double etching step feature good light-trapping and simultaneously good light-coupling behavior. An optimized double-textured (DT) ZnO:Al yielded 12.5% initial efficiency in an a-Si:H/μc-Si:H solar cell with only a 1.64 μm total absorber layer thickness. The described effects on light-coupling and light-trapping based on interface textures and their evolution by absorber layer growth are of general interest and might influence optimization strategies in other photovoltaic technologies. The specificities of the absorption coefficient of the absorber material, layer thicknesses, and the individual growth processes have to be considered.

**Author Contributions:** Conceptualization, G.C.E.J. and J.H.; methodology, J.I.O., T.M. and T.Z.; formal analysis, investigation, visualization, and Writing—Original draft preparation, G.C.E.J.; Writing—Review and editing, supervision, data curation, and funding acquisition, J.H.

**Funding:** This research was funded by BMU (contract No. 03327693A and 0325356B).

**Acknowledgments:** The authors thank H. Siekmann, J. Worbs, S. Bugdol and J. Kirchhoff for technical assistance.

**Conflicts of Interest:** The authors declare no conflict of interest.

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
