# Peer review of "Coupling and Trapping of Light in Thin-Film Solar Cells Using Modulated Interface Textures"

_applsci, doi:10.3390/app9214648_

Round 1

Reviewer 1 Report

The article deals with difficult issues such as the influence of interlayer texture on the optical and electrical parameters of a solar cell. The presented work is at the student's report level and it lacks many results, for example - the absorption coefficient in individual layers.
There is no solar cell scheme with a clear description of the parameters of each functional layer.
Very large periodic oscillations (25-45%) of the reflection coefficient for waves above 800 nm are strange (Fig. 3).
Generally, a lot of technological work but results so-so.

Author Response

Dear Reviewer,

thank you very much for the critical comments. The issue of light management using rough interfaces is well established in thin silicon photovoltaics, while it is quite new to other PV technologies. The manuscript discusses different aspects of light-coupling and light-scattering in solar cells and aims at the transfer of knowledge from silicon thin film PV to other technologies. Your comments raise the problem of the different background knowledge and we will cover these aspects in our revision.

The article deals with difficult issues such as the influence of interlayer texture on the optical and electrical parameters of a solar cell.

In fact we focus on the optical performance only and do not discuss the electrical device performance. We add a comment on this aspect when introducing the double texture: 'Note, that high aspect ratios cause an area increase of the contact interface, which results in stronger interface recombination {Brammer, 2003 #74}, and sharp features may deteriorate the quality of the absorber layer {Python, 2009 #150}. A tradeoff between good optical and good electrical device performance must be found.'

The presented work is at the student's report level and it lacks many results, for example - the absorption coefficient in individual layers. There is no solar cell scheme with a clear description of the parameters of each functional layer.

We now add a graphical abstract illustrating the layer stack, a new Figure 1 with the layer stack and an illustration of the test series discussed in sections 3. In the experimental section we provided several details of the layer thicknesses. Now we added the function of the doped layers and provide typical optical absorption data of individual layers the new Figure 5. Text and captions were adjusted accordingly.

Very large periodic oscillations (25-45%) of the reflection coefficient for waves above 800 nm are strange (Fig. 3).

The low absorption coefficient of silicon in the near infrared spectrum causes interference here. For cells with poor light-scattering these can get that strong. For effective light-scattering and thus light-trapping the interference fringes are strongly reduced. We added a statement about these relations to the text: 'The strong interference fringes in the near-infrared region are caused by the low absorption coefficient of the silicon (below 100 cm-1 at 1000 nm). The amplitude of the interference is reduced for effectively scattering interfaces.'

Generally, a lot of technological work but results so-so.

We hope to make clear now, that we discuss the optical effects of light coupling and scattering in a general way as major part of the story. The technological work is definitely important to achieve those results, especially when optimizing those textures including the double textures.

Reviewer 2 Report

Dear author,

The manuscript “Coupling and trapping of light in Thin-Film Solar Cells using modulated interface textures” gives a clear description and figures of light management to increase solar cell performance. After carefully reading the manuscript, I found it suitable for publishing after some minor corrections, which you can find below.

Sincerely yours,

Reviewer

1) Line 39: Please explain the abbreviation of CIGS

2) The author is strongly recommended to split “Materials and Methods” in subsections. It would make it more overview to read this section.

3) Line 97-98: How was the thickness determine?

4) Line 123: It is not clear to the readers what a-Si:H and µc-Si:H are. Please give a more description about these materials/methods.

5) Line 136: First appearance of PECVD abbreviation. Please add this to line 126.

6) Figure 1 and Figure 2 should be merged in the text of section 3.

7) line 165: Typo of 21.5 Ω/?

8) The title of subsections “single texture” and “double texture” is not clear enough to the readers.

Author Response

Dear Reviewer,
thanks for the valuable comments that helped to improve the manuscript. we adjusted the manuscript accordingly or provide our statement of defense.
1) done incl. specification of perovskite as 'metal halide perovskite'
2) We added new subheadings: ZnO:Al front contacts, Texture etched ZnO:Al, Solar cells. The specific characterization methods are are aligned with the processes of the respective materials/devices
3) added: 'Thicknesses were determined from an etched step using a step profiler without considering volume overestimation of rough surfaces {Owen, 2012 #12}. '
4) We added: ...'hydrogenated amorphous and microcrystalline silicon (a-Si:H/µc-Si:H) tandem solar cells were prepared on these ZnO:Al front contacts. a-Si:H has a bandgap of around 1.7 eV and is used in the top cell, while µc-Si:H with a bandgap close to crystalline silicon (~1.1 eV) is used as the bottom cell absorber. ' However, we do not want to extend the description of the cell preparation further, because it covers already 20 lines of text and we refer to further literature.
5) done
6) We would like to keep the figures as separate figures and split the description in the text as well. Both figures are formally identical with same x and y axes. However, the solar cells are very different (absorber layer thickness, etch solutions resulting in strongly different front contact textures), and thus a quantitative comparison of both sets of data is not valid. In addition, we demonstrate the difference between light-coupling and -scattering in Fig 2, while different light-trapping is discussed in Fig. 3 especially for very large features.
7) The unit Ohm/square is common for sheet resistance indicating the square reference area. However, we used the simplified and physically correct unit Ohm now.
8) We provide new subheadings to clarify the structure of the results section: 3.1: Solar cells on single textured ZnO:Al with different feature size; 3.3: Good coupling and scattering using double textured ZnO:Al